# Effects of mechanical insufflation-exsufflation on ventilator-free days in intensive care unit subjects with sputum retention; a randomized clinical trial

Shota Kubota[1], Hideki Hashimoto[1], Yurika Yoshikawa[2], Kengo Hiwatashi[1], Takahiro Ono[1], Masaki Mochizuki[1], Hiromu Naraba[1], Hidehiko Nakano[1], Yuji Takahashi[1], Tomohiro Sonoo[1], Kensuke Nakamura[1,3]*

1 Department of Emergency and Critical Care Medicine, Hitachi General Hospital, Ibaraki, Japan,
2 Department of Nursing in Emergency and Critical Care Center, Hitachi General Hospital, Ibaraki, Japan,
3 Department of Critical Care Medicine, Yokohama City University Hospital, Kanagawa, Japan

* nakamura.ken.kl@yokohama-cu.ac.jp

**Data Availability Statement:** All relevant data are within the manuscript and its Supporting Information files.

## Abstract

### Background

Mechanical insufflation-exsufflation (MI-E) facilitates extubation. However, its potential to reduce the duration of ventilator use remains unclear. Therefore, the present study investigated whether the use of MI-E shortened the duration of mechanical ventilation in patients with high sputum retention.

### Methods

A randomized open-label trial was conducted at a single intensive care unit (ICU) in Japan between November 2017 and June 2019. Ventilated subjects requiring suctioning at least once every hour due to sputum retention were randomly assigned to the MI-E group or conventional care group. The primary endpoint was the number of ventilator-free days on day 28. Secondary endpoints were ventilator days in surviving subjects, the length of ICU stay, and mortality and tracheostomy rates among survivors.

### Results

Forty-eight subjects (81% males) with a median age of 72 years (interquartile range [IQR], 65–85 years) were enrolled. There were 27 subjects in the MI-E group and 21 in the control group. The median number of ventilator-free days was 21 (IQR, 13–24) and 18, respectively (IQR, 0–23) (P = .38). No significant differences were observed in the ICU length of stay (median, 10 days (IQR, 7–12) vs 12 days (IQR, 6–15); P = .31), mortality rate (19% vs 15%; odds ratio [OR], 1.36 [0.28–6.50]; P = .69), or tracheostomy rate among survivors (14% vs 28%; OR, 0.40 [0.08–1.91]; P = .25).

**Funding:** The author received no specific funding for this work.

**Competing interests:** The authors have declared that no competing interests exist.

## Conclusion

In ventilated subjects in the ICU with high sputum retention, the use of MI-E did not significantly increase the number of ventilator-free days over that with conventional care.

## Introduction

Prolonged mechanical ventilation is associated with mortality and morbidity, including intensive care unit-acquired weakness [1]. Mucociliary clearance is impaired in acutely ill patients [2]. Patients with high sputum retention are at a higher risk of extubation failure than those with low sputum retention [3]. Retained secretions are one of the causes of extubation failure [4]. Endotracheal intubation prevents patients from closing the glottis, which is necessary for effective coughing. In a previous study that investigated cough peak flow, patients with ineffective cough were more likely to experience extubation failure [5].

Mechanical insufflation-exsufflation (MI-E) is a device that promotes the removal of secretions and attenuates atelectasis by alternately applying positive and negative pressures throughout the airways [6]. Endotracheal tube suctioning is difficult to expectorate in the left bronchus, while MI-E induces similar air flow in both airways [4]. A recent study reported that in adult ICU subjects receiving mechanical ventilation for more than 24 hours, the weight of aspirated airway mucus and lung compliance values were higher in the group receiving respiratory physiotherapy with an MI-E device than in the group receiving standard respiratory physiotherapy alone [7]. Another study suggested that the use of MI-E followed by endotracheal suctioning reduced peak airway pressure and airway resistance, and increased lung compliance over that with isolated endotracheal suctioning in subjects on mechanical ventilation [8].

Observational trials compared the usefulness of MI-E for ventilated subjects: the use of MI-E on ventilated subjects with neuromuscular diseases facilitated weaning off the ventilator and prevented reintubation [9,10]. However, the types of patients who will benefit from MI-E and whether the use of MI-E during mechanical ventilation reduces the number of ventilator days remain unclear. Therefore, the present study investigated whether the additional use of MI-E in conventional pulmonary care shortened the duration of mechanical ventilation in subjects with high airway secretions. The hypothesis of the present study was that the additional use of MI-E in conventional pulmonary care shortens the duration of mechanical ventilation in subjects with high airway secretions.

## Methods

### Study design and participants

The present study was conducted according to the CONSORT statement (S1 File). It was an open-label, single-center, randomized, controlled trial conducted in the ICU of Hitachi General Hospital, a tertiary care center in Japan. This ICU is a medical and surgical unit for patients admitted from the Emergency department and those who deteriorate rapidly during hospitalization. Patients admitted to the ICU between 1st November 2017 and 30th June 2019 were eligible for the present study if they met all of the following criteria: (1) being intubated and 18 years of age or older; (2) requiring sputum suctioning at least once every hour due to a large amount of sputum; (3) receiving ventilator management for more than 24 hours before inclusion; (4) expected to require ventilator management for more than 48 hours. Patients

were included if they required suctioning more than once every hour for the first 24 hours after intubation. If they did not require suctioning once every hour in the initial 24 hours, but needed suctioning at least once every hour in the subsequent 24 consecutive hours, they were included at that point. There was no specific time limit on inclusion regarding the number of days elapsed after intubation. Patients who met the following criteria were excluded from the study: pregnant women, individuals with intracranial hypertension, patients with neurological conditions that make spontaneous breathing difficult, individuals unable to provide consent for the study, patients unwilling to agree to the use of full life support, and patients considered by the attending physician to have a coexisting condition that is exacerbated by MI-E, such as pneumothorax, empyema with fistula, or hemoptysis (S2 and S3 Files).

## Ethics approval and consent to participate

The present study was conducted in accordance with the Declaration of Helsinki. It was approved by the Ethics Review Board of Hitachi General Hospital (Number 2018–66) (S4 and S5 Files) and registered in the University Hospital Medical Information Network (registration number UMIN000033980). Written informed consent to participate in the present study was obtained from all subjects or their legal representatives. Since the present study involved intubated patients, written informed consent was obtained from their legal representatives at the time of inclusion for the majority of patients. Whenever possible, informed consent was also obtained from patients after extubation.

## Randomization and procedures

Subjects were randomly assigned by attending physicians to the MI-E or control group with a 1:1 ratio. Included subjects were assigned a random number of 0 or 1 using software (File-Maker pro 16; FileMaker, Inc., Santa Clara, CA, USA), designating them as members of the control or MI-E group, respectively. Simple randomization was implemented and did not include random blocking.

In the control group, all subjects received standard medical therapy, which included supplemental oxygen, respiratory physiotherapy, bronchodilators, antibiotics, ventilator management, weaning from the ventilator, and any other therapies selected by the ICU attending physician and ICU staff. Daily assessments of the ability to breathe with pressure support ventilation were performed when the positive end-expiratory pressure and fraction of inspiratory oxygen levels were lower than the day before. Additionally, the ventilator was switched to pressure support ventilation if the attending physician considered the patient sufficiently awake to breathe with pressure support ventilation. Extubation was attempted in patients with a successful spontaneous breathing trial (SBT). The ICU attending physician made the final decision regarding whether to extubate the patient. Tracheostomy was performed if the ventilator duration was expected to be more than 14 days. Airway secretions were cleared through endotracheal tube suctioning with appropriate humidification for airway clearance. Suctioning was performed on-demand by the nurse in charge who fully understood the protocol of the study and had received adequate training and practical experience with suctioning. Although subjects who required suctioning at least once every hour at the start of the study were included in the research, there was no upper limit to the number of times. Moreover, if it was not possible to suction airway secretions after inclusion and no further suctioning of airway secretions was expected to be possible based on a physical examination and auscultation, suctioning was allowed to occur less frequently than once an hour. In addition to the treatments employed in the control group, the MI-E COMFORT COUGH- (Kahubentekku Corporation, Japan) [11] was additionally used in the MI-E group prior to each suctioning from randomization until

extubation. When the MI-E COMFORT COUGH- is directly connected to the endotracheal tube and the pressure is set to 40 cmH$_2$O, it applies both positive and negative pressures of 40 cmH$_2$O during inhalation and exhalation, respectively, inside the endotracheal tube. MI-E was used 10 times for each sputum aspiration with a pressure of 40 cmH$_2$O for insufflation and -40 cmH$_2$O for exsufflation through the endotracheal tube. In the MI-E group, if there were no more airway secretions to be cleared even after using MI-E, suctioning and the use of MI-E were permitted at intervals exceeding one hour. Humidification was performed by connecting a heated humidifier to the continuous tube from the endotracheal tube. When the nurse considered the use of a nebulizer to be more effective at facilitating the expectoration of airway secretions, the nebulizer was used on-demand. Authors had access to information that identified individual participants during and after data collection.

## Outcomes

The primary outcome was the number of ventilator-free days by day 28. Ventilator-free days were defined as the number of days a subject was alive and free from mechanical ventilation for at least 24 consecutive hours between randomization and day 28. In subjects extubated once and reintubated, every 24 hours without a ventilator was calculated as one day and added to the number of ventilator-free days. In subjects without tracheostomy, a period of at least 24 consecutive hours without reintubation after extubation was defined as one ventilator-free day. In subjects with tracheostomy, breathing without ventilatory assistance for at least 24 consecutive hours was defined as one ventilator-free day. In the present study, we considered breathing through an anatomical airway and breathing through tracheostomy to be comparable in terms of their ability to remain free from mechanical ventilation for 24 hours. Secondary outcomes were the duration of mechanical ventilation up to day 28, the ICU length of stay, the mortality rate, and the tracheostomy rate. Secondary outcomes, except for mortality, were analyzed exclusively for patients who survived. Baseline characteristics, such as age, sex, the Acute Physiology and Chronic Health Evaluation (APACHE II) score, PaO$_2$/FiO$_2$ ratio, C-reactive protein, procalcitonin, and the reason for ICU admission were reported as numbers and percentages or as medians and interquartile ranges (IQRs). Safety and adverse events were monitored by the investigator during the present study through a clinical examination, vital signs, and laboratory investigations. Adverse events were noted by a retrospective review of medical records and not taken systematically in this study.

## Statistical analysis

The sample size was obtained by a calculation using the following method. In a previous study performed in Hitachi General Hospital, the standard deviation of ventilator-free days up to 28 days for subjects ventilated for more than 3 days in the ICU was 7.6 days [12]. Since the standard deviation was expected to decrease when subjects were limited to those with high sputum retention, we assumed the baseline standard deviation of ventilator-free days in the present study to be 5 days. A sample size of 25 subjects per group was estimated to provide 80% power to detect a reduction of 4 ventilator-free days in the MI-E group for the primary endpoint, with a two-sided 5% type 1 error rate. To account for withdrawals during the present study, we set the sample size of participants as 30 per group. We planned to terminate enrollment once we reached the target sample size or in June 2019. The analysis plan was finalized on 16th October 2017, prior to the recruitment of the first participant on 1st November 2017. In the present study, we performed an intention-to-treat analysis.

To analyze the primary outcome and secondary outcomes, except for mortality and tracheostomy rates, the Mann-Whitney U test was used with 95% CIs for superiority. To analyze

mortality and tracheostomy rates as well as subject characteristics, the chi-square test or Fisher's exact test were used where appropriate. The duration of mechanical ventilation up to day 28 in surviving subjects was compared using Kaplan-Meier survival curves. Survival times were calculated from the time of randomization until the time of death or loss to the follow-up. All statistical tests were two-sided, with a significance level of p <0.05. All analyses were performed using R software, version 2.7–1.

## Results

Enrollment concluded in accordance with the pre-established schedule, culminating in June 2019. Between November 2017 and June 2019, 514 patients received ventilator management in the Hitachi General Hospital ICU. Forty-eight subjects met the inclusion criteria and were subsequently enrolled in the present study. Among the 48 enrolled subjects, 27 were assigned to the MI-E group and 21 to the control group (Fig 1). The protocol was strictly adhered to in both groups and no subjects withdrew from the study. There were no missing data (S1 Checklist).

No significant differences were observed in baseline characteristics between the groups (Table 1). The median P/F ratio for both groups was less than 200 mmHg. More than 80% of subjects were admitted to the ICU for medical reasons. None of the subjects died within 24 hours of the intervention.

The median number of ventilator-free days by day 28 after randomization did not significantly differ between the MI-E and control groups (21 days, IQR, 13–24 vs 18 days, IQR, 0–23; P = 0.38) (Table 2). The median duration of mechanical ventilation in surviving subjects was 6 (IQR, 3–9) in the MI-E group and 8 (IQR, 4–14) in the control group; P = 0.19. The median ICU length of stay was 10 (IQR, 7–12) in the MI-E group and 12 (IQR, 6–15) in the control

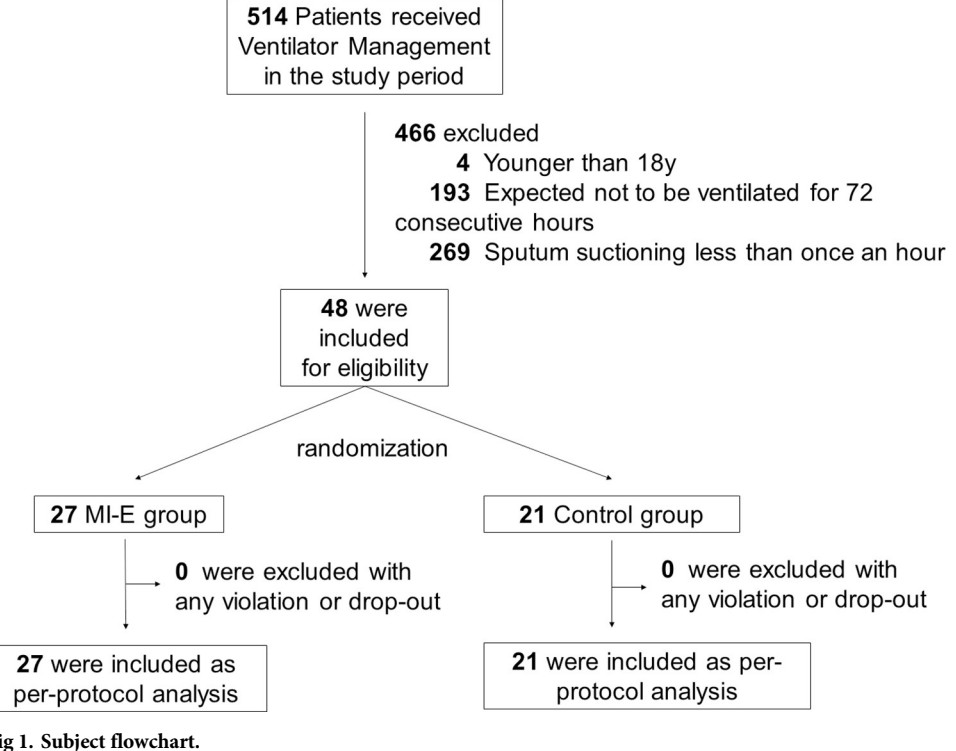

**Fig 1. Subject flowchart.**

**Table 1. Subject characteristics.**

| Characteristics | | Control group (n = 21) | MI-E group (n = 27) |
|---|---|---|---|
| Age, median (IQR), years | | 71 (58–81) | 73 (67–86) |
| Age-category, no. (%) | ≤ 60 | 6 (28.6) | 4 (14.8) |
| | 61–80 | 9 (42.9) | 11 (40.7) |
| | ≥ 81 | 6 (28.6) | 12 (44.4) |
| Male subjects, no. (%) | | 17 (81.0) | 22 (81.5) |
| APACHE II score, median (IQR) | | 16 (13–23) | 19 (14–27) |
| PaO$_2$/FiO$_2$, median (IQR), mmHg | | 181 (138–264) | 199 (173–258) |
| CRP, median (IQR), mg/dL | | 12 (3–16) | 12 (5–17) |
| PCT, median (IQR), ng/mL | | 2.9 (0.6–7.2) | 1.6 (0.3–4.3) |
| History of smoking, no. (%) | | 7 (33.3) | 13 (48.1) |
| Bullae, no. (%) | | 13 (61.9) | 16 (59.3) |
| Maximum size of the bullae, median (IQR), mm | | 12 (5–17) | 12 (7–30) |
| Reason for ICU admission, no. (%) | | | |
| Medical | | 18 (85.7) | 23 (85.2) |
| Surgical | | 3 (14.3) | 4 (14.8) |
| Reason for mechanical ventilation, no. (%) | | | |
| Pneumonia | | 12 (57.1) | 15 (55.6) |
| Cardiac arrest | | 2 (9.5) | 4 (14.8) |
| Sepsis | | 3 (14.3) | 1 (3.7) |
| Cerebrovascular diseases | | 2 (9.5) | 2 (7.4) |
| Postoperative ventilation | | 1 (4.8) | 2 (7.4) |
| Heart failure | | 0 (0) | 1 (3.7) |
| Pulmonary embolism | | 0 (0) | 1 (3.7) |
| Acute pancreatitis | | 0 (0) | 1 (3.7) |
| Hemoptysis | | 1 (4.8) | 0 (0) |

Abbreviations: MI-E, mechanical insufflation-exsufflation; IQR, interquartile range; APACHE II score, acute physiology and chronic health evaluation II score; CRP, C-reactive protein; PCT, procalcitonin; ICU, intensive care unit

group; P = 0.31. The 28-day mortality rate was not significantly different between the two groups: P = 0.71. The tracheostomy rate among surviving subjects did not significantly differ between the two groups; P = 0.43 (Table 2). The median duration from intubation to

**Table 2. Clinical outcomes of subjects in MI-E and control groups.**

| Results | Control (n = 21) | MI-E (n = 27) | p-value |
|---|---|---|---|
| **Primary outcome** | | | |
| **Ventilator-free days by day 28, median (IQR)** | 18 (0–23) | 21 (13–24) | 0.38 |
| **Secondary outcomes** | | | |
| duration of mechanical ventilation in surviving subjects, median (IQR) | 8 (4–14) | 6 (3–9) | 0.19 |
| ICU length of stay in surviving subjects, median (IQR) | 12 (6–15) | 10 (7–12) | 0.31 |
| 28-day mortality, no./total (%) | 3/21 (14.2) | 6/27 (22.2) | 0.71 |
| Tracheostomy in surviving subjects, no./total (%) | 4/18 (22.2) | 3/21 (14.3) | 0.68 |

Abbreviations: MI-E, Mechanical insufflation-exsufflation; IQR, interquartile range; ICU, intensive care unit.

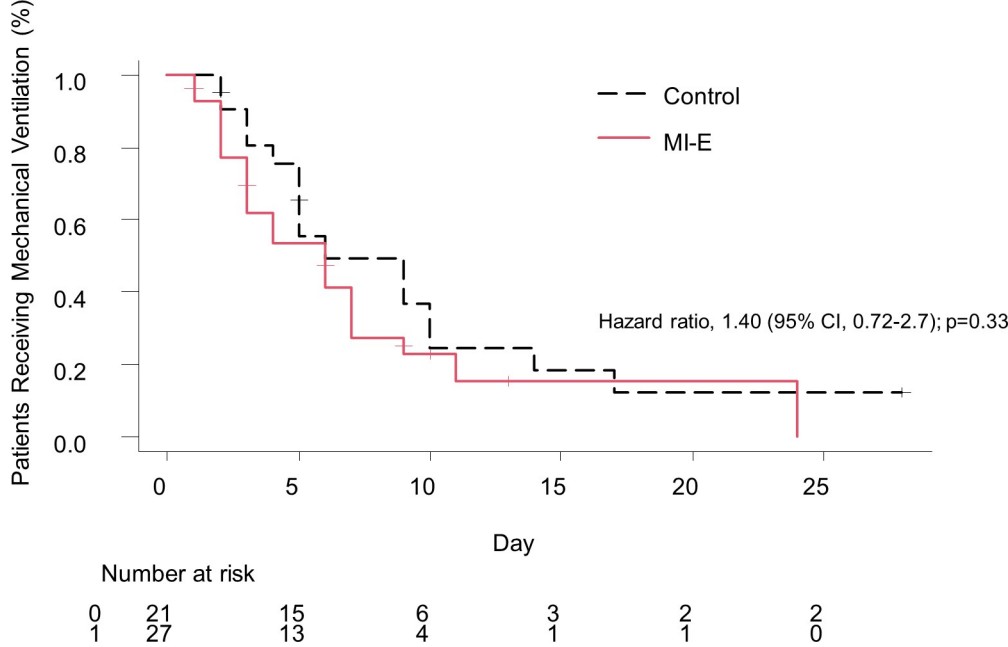

**Fig 2. Kaplan–Meier analysis of the duration of mechanical ventilation.**

tracheostomy was 12 (IQR, 9–19) in the MI-E group and 12 (IQR, 6–15) in the control group; P = 0.85. The Kaplan–Meier analysis of the duration of mechanical ventilation up to 28 days in surviving subjects showed no significant difference; hazard ratio, 1.40 (95% CI, 0.72–2.7); p = 0.33 (Fig 2). None of the outcomes significantly differed. While some patients had hemodynamic instability or arrhythmia at the time of inclusion, no new cases of hemodynamic instability or arrhythmia occurred with the use of MI-E. The use of MI-E did not lead to new cases of lung injury due to barotrauma, volutrauma, atelectrauma, or biotrauma. No other apparent adverse effects attributed to MI-E occurred in the present study, such as pneumothorax, hemoptysis, oxygen desaturation, and mucus plugging.

## Discussion

The hypothesis of the present study was that the additional use of MI-E in conventional pulmonary care shortens the duration of mechanical ventilation in subjects with high airway secretions. However, the use of MI-E before each direct tracheal suction did not significantly reduce the number of ventilator-free days by day 28. In addition, no significant differences were observed in the duration of mechanical ventilation, the ICU length of stay, the mortality rate, or the tracheostomy rate between the MI-E and control groups.

Several factors may account for the observed lack of a significant increase in the number of ventilator-free days with MI-E. The primary contributing factor is the limited sample size. Prior to the initiation of the present study, the target difference in ventilator-free days between the two groups was designed as 4 days and the sample size was set at 60 subjects. We planned to terminate enrollment once the target sample size was reached or in June 2019. Fewer ICU-ventilated subjects met the inclusion criteria than expected, and, thus, the study was terminated before the target sample size was reached. The insufficient sample size may have affected the power of the study. Therefore, randomized open-label trials with larger sample sizes are needed. Furthermore, the percentage of patients with a history of smoking may have affected

the results. In the present study, 33% of patients in the control group and 48% in the MI-E group had a history of smoking. Since smoking is known to increase the amount of airway mucus [13], the higher percentage of patients with a smoking history in the MI-E group than in the control group may also have affected the results. Patients in the control group were younger than those in the MI-E group, which may have affected the results obtained. In addition, medical factors may have contributed to the lack of a significant difference. In patients without neuromuscular disease, even if airway secretions are high, the ability to exhale may be sufficient without MI-E. While the use of MI-E may reduce airway resistance or enhance compliance [7,8], it does not directly address the underlying medical factors (such as pneumonia, cardiac arrest, sepsis, or surgery) that necessitated intubation. This may explain why the number of ventilator-free days was not significantly different. Finally, the possibility that the pressure settings for MI-E were inappropriate may be a reason for the lack of a significant difference. A pressure of ±40 cmH2O may have been insufficient for patients with excessive sputum production, such as those examined in the present study. On the other hand, although MI-E was effective, potential adverse events, such as barotrauma or atelectrauma, may have affected the results obtained.

To the best of our knowledge, this was the first randomized controlled trial to directly assess the efficacy of MI-E at shortening the duration of ventilator management or improving other clinical patient-oriented outcomes. A previous study indicated that sufficient humidification and as-needed suctioning serve as the basis for secretion management in mechanically ventilated patients [14], and the present study examined the impact of incorporating MI-E on the number of ventilator-free days. Observational trials compared the usefulness of MI-E for ventilated subjects: the use of MI-E on ventilated subjects with neuromuscular diseases was reported to facilitate weaning off the ventilator and prevented reintubation [9,10]. MI-E is increasingly being utilized in patients with neuromuscular diseases to mitigate pulmonary congestion and reduce the risk of respiratory tract infections [15]. However, due to the distinctive nature of these subjects, these findings cannot be applied to the general ventilated population. In another study involving ventilated subjects with a successful SBT, MI-E performed three times per day after extubation significantly reduced the rate of reintubation to lower than that with conventional pulmonary care [16]. However, MI-E was not used before successful SBT. In the present study, we attempted to demonstrate that the use of MI-E during ventilator management before SBT contributed to better clinical patient-oriented outcomes, such as ventilator-free days and mortality, than those with conventional pulmonary care in a population not limited to those with neuromuscular disease. However, MI-E may cause adverse events. In the present study, MI-E gives a pressure of 40 cmH$_2$O for insufflation and -40 cmH$_2$O for exsufflation through the endotracheal tube. Although a previous study that investigated the use of MI-E in healthy Landrace-Large White female pigs showed that +40/-70 cmH$_2$O was the most effective pressure combination [17], MI-E settings with a pressure of ±40 cmH$_2$O are most commonly used in invasively ventilated critically ill patients [18]. Previous studies demonstrated the safety of MI-E; [19,20] however, conflicting findings have been reported [21]. Since another study demonstrated that pressures of ±54 cmH$_2$O were tolerated well [10], we assumed that a pressure of ±40 cmH$_2$O may also be tolerated. A high positive pressure may increase the risk of barotrauma [22,23]. Furthermore, a high negative pressure may augment lung injury [24]. A previous study suggested that the use of MI-E increased the incidence of chest pain [25]. In 13 studies that used MI-E, 10 found no adverse events, while 3 reported oxygen desaturation, hemodynamic variation, pneumothorax, mucus plugging, hemoptysis, and chest pain as adverse events [18]. In the present study, 33% of patients in the control group and 48% in the MI-E group had a history of smoking. All patients had undergone chest CT, and the percentage of patients with bullae in the lungs was 62% in the control

group and 59% in the MI-E group. The median of the maximum size of bullae was 12 mm (IQR, 5–17) in the control group and 12 mm (IQR, 7–30) in the MI-E group. Pneumothorax did not occur in patients, even in those with a history of smoking or bullae. Pneumothorax did not occur in patients, even in those with a history of smoking or bullae. MI-E may be effective for subjects with high sputum retention, but may also exert negative effects due to the high positive or negative pressure.

The present study has several limitations. Since blinding was not possible due to the nature of the intervention, there may have been performance, reporting, and selection biases. Furthermore, not all nurses had received uniform training in suctioning; therefore, there was a potential for selection bias. The timings of suctioning and nebulization were selected by nurses based on a comprehensive assessment, including physical and auscultation findings and whether airway secretions had been cleared during the previous suctioning. However, due to the absence of standardized rules for the timing of suctioning and nebulization, there was a potential for selection bias. Suctioning may have been performed more effectively and carefully in the MI-E group. MI-E increased the amount of secretions, which may have led to the attending physician delaying extubation due to excessive secretions. Furthermore, since this was a single-center study with a small sample size, generalizability may be lacking. Moreover, it was not possible to separate the effective and harmful effects of MI-E. Although a retrospective review of medical records revealed no adverse events, adverse events were not systematically examined in the present study; therefore, we cannot exclude potential damage, such as barotrauma and atelectrauma. The optimal frequency of MI-E usage remains unclear, and some studies used MI-E 3 to 5 times [4,26]. The utilization of MI-E 10 times may have increased the risk of potential side effects. The number of seconds that MI-E was used for each inhalation and exhalation was also not analyzed in the present study. These parameters may have affected the peak expiratory flow and outcomes of this study. Additionally, in both groups, subjects who required suctioning at least once every hour at the start of the study were included in the research. However, if they no longer needed frequent suctioning based on the clinical assessment after study inclusion, suctioning less than once every hour was allowed. It was difficult to establish a fixed number of suctions per hour because additional suctioning or the use of MI-E in the absence of further cleared airway secretions may have led to potential side effects without corresponding benefits. However, variability in the frequency of suctioning and MI-E sessions that individual patients received each day may have affected the outcomes. Suction frequency was initially low immediately after intubation; however, patients were included if they required suctioning at a rate of once per hour or more within the subsequent continuous 24-hour period. This variation in the time between intubation and inclusion may have potentially affected the results obtained herein. Regarding the exclusion criteria of the present study, patients considered by the attending physician to have a coexisting condition exacerbated by MI-E, such as pneumothorax, empyema with fistula, or hemoptysis, were excluded. However, since the study protocol did not provide a detailed definition of exclusion criteria, there was a potential for selection bias. The number of patients in both groups was not 1:1, which may have affected the results obtained because simple randomization was implemented and did not include random blocking. In the present study, the sample size was calculated to detect a reduction of 4 ventilator-free days in the MI-E group for the primary endpoint. Since the present results showed that the duration of mechanical ventilation was 6 in the MI-E group and 8 in the control group, setting the cut-off of the primary outcome at 4 days may have been too large. This could have potentially influenced the results. The period during which suctioning was performed at least once every hour was not systematically recorded in the present study. Furthermore, there was no systematic record of the medical therapies administered, apart from MI-E, in the present study. The lack of systematic data

collection may have had an impact on study outcomes. In the present study, ventilator-free days for patients following extubation and tracheotomy without mechanical ventilation were counted as equivalent. While tracheotomized patients are more likely to clear airway secretions than extubated patients, there is no feasible method to unify or convert ventilator-free days between extubated and tracheotomized patients. Therefore, we were compelled to treat ventilator-free days in extubated and tracheotomized patients as equivalent. The patients included in this study were those who required suctioning at least once every hour. Therefore, it is important to note that the results and conclusions drawn from this study may not be applicable or generalizable to all patients who are on ventilators.

## Conclusions

In ventilated subjects in the ICU with high sputum retention, the use of MI-E did not significantly increase the number of ventilator-free days over that with conventional care.

## Supporting information

**S1 Checklist.**
(DOCX)

**S1 File. Data pertaining to this research investigation.**
(PDF)

**S2 File. The study protocol in English.**
(PDF)

**S3 File. The study protocol in Japanese.**
(DOCX)

**S4 File. The original approval document in English.**
(PDF)

**S5 File. The original approval document in Japanese.**
(XLSX)

## Acknowledgments

There are no acknowledgments for this work.

## Author Contributions

**Conceptualization:** Shota Kubota.

**Data curation:** Shota Kubota, Hideki Hashimoto, Yurika Yoshikawa, Kengo Hiwatashi, Takahiro Ono, Masaki Mochizuki, Hiromu Naraba, Hidehiko Nakano, Yuji Takahashi, Tomohiro Sonoo, Kensuke Nakamura.

**Formal analysis:** Shota Kubota.

**Methodology:** Shota Kubota.

**Project administration:** Shota Kubota, Hideki Hashimoto, Yurika Yoshikawa, Kengo Hiwatashi, Takahiro Ono, Masaki Mochizuki, Hiromu Naraba, Hidehiko Nakano, Yuji Takahashi, Tomohiro Sonoo, Kensuke Nakamura.

**Resources:** Shota Kubota, Hideki Hashimoto, Yurika Yoshikawa, Kengo Hiwatashi, Takahiro Ono, Masaki Mochizuki, Hiromu Naraba, Hidehiko Nakano, Yuji Takahashi, Tomohiro Sonoo, Kensuke Nakamura.

**Supervision:** Hideki Hashimoto.

**Writing – original draft:** Shota Kubota.

**Writing – review & editing:** Hideki Hashimoto.

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
