## [Decision Letter · Decision Letter 0]

6 Oct 2023

PONE-D-23-25920Effects of mechanical insufflation-exsufflation on ventilator-free days in intensive care unit subjects with sputum retention; A randomized clinical trialPLOS ONE

Dear Dr. Nakamura,

Thank you for submitting your manuscript to PLOS ONE. After careful consideration, we feel that it has merit but does not fully meet PLOS ONE’s publication criteria as it currently stands. Therefore, we invite you to submit a revised version of the manuscript that addresses the points raised during the review process.

We look forward to receiving your revised manuscript.

Kind regards,

Jean Baptiste Lascarrou

Academic Editor

PLOS ONE

Journal Requirements:

**Additional Editor Comments:**

- Please revise the manuscript according to reviewer's comments who raised concerns about structure and interpretation of the results. Be aware, than significant proportion of manuscripts submitted as revised version could be rejected.- Revised version must be presented in an intelligible fashion and written in standard English ("No" for 3 reviewers).

Reviewers' comments:

Reviewer's Responses to Questions

**Comments to the Author**

1. Is the manuscript technically sound, and do the data support the conclusions?

Reviewer #1: No

Reviewer #2: Yes

Reviewer #3: Partly

2. Has the statistical analysis been performed appropriately and rigorously? 

Reviewer #1: Yes

Reviewer #2: Yes

Reviewer #3: Yes

3. Have the authors made all data underlying the findings in their manuscript fully available?

Reviewer #1: Yes

Reviewer #2: Yes

Reviewer #3: Yes

4. Is the manuscript presented in an intelligible fashion and written in standard English?

Reviewer #1: No

Reviewer #2: No

Reviewer #3: No

5. Review Comments to the Author

Reviewer #1: In this prospective, single-center, randomized trial, Kubota and colleagues compared the effects of mechanical insufflation-exsufflation on ventilator-free days among 48 patients ventilated for more than 24h. They major findings are that outcomes did not differ between the 2 groups. Although the authors should be commended for conducting such a trial, we can regret that it was prematurely stopped, leading to a lack of statistical power. Please find below my comments to try to improve the manuscript.

The introduction should better reflect the current knowledge on the topic. Reporting results from observational studies in intubated patients in which mechanical insufflation-exsufflation was tested to hasten weaning from mechanical ventilation (ref 14 and 15) may be more appropriate than reporting the results from Gonçalves and colleagues who tested mechanical insufflation-exsufflation after extubation (not “peri-extubation” as stated line 68) to avoid reintubation.

I have major concerns regarding the design of the trial:

• What is the airway suctioning protocol in the unit? I it “on-demand” or “scheduled?

• For how many hours should airway be suctioned more than once per hour before inclusion?

• What was the time limit for inclusion? Could a patient be included after 10 days of mechanical ventilation, which may have biased the results?

• Was there a standardized weaning protocol? In the absence of a weaning protocol, mechanical insufflation-exsufflation may have delayed extubation. Indeed, mechanical insufflation-exsufflation increases the amount of secretions which may have led the attending physician to delay extubation due to excessive secretions.

• What is the tracheostomy policy? Is it early systematic tracheostomy, or late tracheostomy in selected patients?

• Who performed the mechanical insufflation-exsufflation? Please detail the protocol (the sentence “mechanical insufflation-exsufflation was used 10 time for each sputum aspiration”, lines 112-113 is not clear), including its weaning (line 116 is not clear).

• The secondary outcomes should be listed explicitly.

• Respectfully, “the ability to remain free from mechanical ventilation for 24 hours” is not the same between extubation and tracheostomy (lines 130-132). Airway suctioning is easier in tracheostomized patients than in extubated patients.

• Please detail the adverse events reviewed.

• For statistical analysis, please specify the expected number of ventilator free days, in the control group to better understand whether the sample size calculation was realistic.

• For primary outcome, it is not clear which statistical test was used (Mann-Whitney? Cox model?). Additionally, the statistical analysis paragraph can easily be shortened and simplified to be more readable.

The results are overinterpreted. The ventilator-free days are not “slightly higher” in the mechanical insufflation-exsufflation, they were not different. “Secondary outcomes […] were better in the mechanical insufflation-exsufflation group” is not true, they were not different. Additionally, the tracheostomy rate among survivors is not clear. According to Table 2, in the intervention group, 6 out of 27 patients died, so 21 patients survived. Therefore, the tracheostomy rate should be 3/21, not 3/22.

In the discussion section, please also discuss the fact that the absence of difference could be due to the population selected which could not be the good target, or to the absence of efficacy of the device.

Does the “the number of days of ventilatory management” means “duration of mechanical ventilation”?

English editing would greatly improve the reading of the manuscript.

Reviewer #2: This study is one of the few RCT studies in the intensive care field of M I-E and the results are scientific importance.

Several matters are commented on below that require mentioning.

1. The exclusion criteria do not contain pneumothorax, which is the main adverse event when using M I-E.

2. More than 80% of the subjects are male, but how many have a smoking history or chest C T that may have a bullae?　The MI-E pressure of 40 cm H2O was performed in all the same cases, but it is necessary to mention how it was managed and whether there were no adverse events.

3. In line 106 of the Randomisation and procedures, the word "respiratory chest physiotherapy" are not often used and should be corrected to either "chest physiotherapy" or "respiratory physiotherapy".

4. Lines 227-229; the reference to 40 mmHg is cited as justification for setting MI-E pressure in this study, but the unit of 40 mmHg is 54 cm H2O when calculated in cm H2O, which may mislead the reader. Reference: https://pubmed.ncbi.nlm.nih.gov/25492956/

5. The structure of the discussion lacks logic.

It would be better to describe what was expected in this hypothesis, what were the results, what were the factors, sample size and other factors due to the study design have been mentioned, but are there other factors, are there any respiratory physiological factors, and should include reference to medical factors.

Reviewer #3: This is an interesting study that could be improved by addressing the added comments.

6. PLOS authors have the option to publish the peer review history of their article (what does this mean?). If published, this will include your full peer review and any attached files.

Reviewer #1: No

Reviewer #2: No

Reviewer #3: **Yes: **Willemke Stilma

---

## [Author Response · Author response to Decision Letter 0]

18 Nov 2023

Dear Dr. Lascarrou,

Thank you for your e-mail regarding our manuscript entitled “Effects of mechanical insufflation-exsufflation on ventilator-free days in intensive care unit subjects with sputum retention; A randomized clinical trial”, and the valuable comments of the three Reviewers. Please find attached our revised manuscript as well as a point-by-point responses to the Reviewers’ comments. We apologize for the initial submission, which was not written in standard English. The manuscript was proofread by a native English speaker to enhance its readability. I have received the following comment from Reviewer 3: "The email address of the corresponding author does not provide any information on who this is or what institution is behind this. Could this be checked and changed into a professional email address?" We apologize for the non-formal e-mail address, which was changed to a professional e-mail address in the revised manuscript to clarify the affiliation. Furthermore, since the corresponding author is affiliated with two institutions, his affiliation was updated to reflect this. We changed the following e-mail address from “mamashockpapashock@yahoo.co.jp” to “nakamura.ken.kl@yokohama-cu.ac.jp” We also added the following as affiliations:“Department of Critical Care Medicine, Yokohama City University Hospital, Kanagawa, Japan” PACE was used to ensure that the figures complied with PLOS ONE’s submission guidelines.

We consider the revised manuscript to be a suitable response to the Reviewers’ comments and it has been significantly improved over the initial submission. We trust that it is now suitable for publication in PLOS ONE. Thank you in advance for your kind consideration of this manuscript.

Sincerely yours,

Shota Kubota

Department of Emergency and Critical Care Medicine, Hitachi General Hospital, Ibaraki, Japan

RESPONSE TO REVIEWER 1:

We wish to express our strong appreciation to the Reviewer for their insightful comments, which have helped us to significantly improve the quality of the revised manuscript.

Comment 1: The introduction should better reflect the current knowledge on the topic. Reporting results from observational studies in intubated patients in which mechanical insufflation-exsufflation was tested to hasten weaning from mechanical ventilation (ref 14 and 15) may be more appropriate than reporting the results from Gonçalves and colleagues who tested mechanical insufflation-exsufflation after extubation (not “peri-extubation” as stated line 68) to avoid reintubation.

Response: We appreciate the Reviewer’s comment. Studies that used mechanical insufflation-exsufflation to hasten weaning from mechanical ventilation in intubated patients were cited in the revised manuscript instead of those that used mechanical insufflation-exsufflation after extubation. 

In accordance with the Reviewer’s comment, we changed the following text from 

“Previous studies demonstrated the efficacy of MI-E in extubation failure in critical care settings. Gonalves MR et al. reported that the peri-extubation use of MI-E increased the probability of successful extubation.”

to 

“Observational trials compared the usefulness of MI-E for ventilated subjects: the use of MI-E on ventilated subjects with neuromuscular diseases facilitated weaning off the ventilator and prevented reintubation.” (lines 69-71)

Moreover, in accordance with the Reviewer’s comment, when referring to the findings reported by Gonçalves and colleagues, the expression “peri-extubation” was changed to “after extubation”. We apologize for the ambiguous wording.

Comment 2: What is the airway suctioning protocol in the unit? I it “on-demand” or “scheduled?

Response: We wish to thank the Reviewer for this comment. We agree that more precise details were needed on the frequency and method of suction. Suctioning was performed on-demand. Although subjects who required suctioning at least once every hour at the start of the study were included in the research, there was no upper limit to the number of times. Moreover, if it was not possible to suction airway secretions after inclusion, and no further suctioning of airway secretions was expected to be possible based on a physical examination and auscultation, suctioning was allowed to occur less frequently than once an hour. This may have affected the results obtained. However, performing additional suctioning on patients who do not produce further airway secretions despite suctioning may result in more side effects than benefits; therefore, we were unable to maintain the same frequency for all patients. We documented this as a limitation in the Discussion section.

In accordance with the Reviewer’s comment, we changed the following sentences from (lines 116-118)

“However, if they no longer needed to suction frequently based on the clinical assessment after study inclusion, they were allowed to suction less often than once every hour.”

to

“Suctioning was performed on-demand by the nurse in charge who fully understood the protocol of the study and had received adequate training and practical experience with suctioning. Although subjects who required suctioning at least once every hour at the start of the study were included in the research, there was no upper limit to the number of times. Moreover, if it was not possible to suction airway secretions after inclusion and no further suctioning of airway secretions was expected to be possible based on a physical examination and auscultation, suctioning was allowed to occur less frequently than once an hour.” (line 131-139)

and

“In the MI-E group, if there were no more airway secretions to be cleared even after using MI-E, suctioning and the use of MI-E were permitted at intervals exceeding one hour.” (line 146-148)

Furthermore, the following sentences were added to the Discussion section.

“It was difficult to establish a fixed number of suctions per hour because additional suctioning or the use of MI-E in the absence of further cleared airway secretions may have led to potential side effects without corresponding benefits. However, variability in the frequency of suctioning and MI-E sessions that individual patients received each day may have affected the outcomes.” (line 334-339)

Comment 3: For how many hours should airway be suctioned more than once per hour before inclusion? What was the time limit for inclusion? Could a patient be included after 10 days of mechanical ventilation, which may have biased the results?

Response: We wish to thank the Reviewer for this comment. We apologize for the unclear description of patient inclusion criteria. To clarify which patients were included and at what specific times, the following sentences were added to the Methods section.

“Patients were included if they required suctioning more than once every hour for the first 24 hours after intubation. If they did not require suctioning once every hour in the initial 24 hours, but needed suctioning at least once every hour in the subsequent 24 consecutive hours, they were included at that point. There was no specific time limit on inclusion regarding the number of days elapsed after intubation.” (line 91-96)

As suggested, there may be a potential for bias due to variations in the period between intubation and inclusion. Therefore, the following sentences were added to the Discussion section.

“Suction frequency was initially low immediately after intubation; however, patients were included if they required suctioning at a rate of once per hour or more within the subsequent continuous 24-hour period. This variation in the time between intubation and inclusion may have potentially affected the results obtained herein.” (line 339-342)

Comment 4: Was there a standardized weaning protocol? In the absence of a weaning protocol, mechanical insufflation-exsufflation may have delayed extubation. Indeed, mechanical insufflation-exsufflation increases the amount of secretions which may have led the attending physician to delay extubation due to excessive secretions.

Response: We appreciate the Reviewer’s comment on this point. In the present study, there was no specific weaning protocol defined in the research protocol. Weaning decisions were made at the discretion of the attending physician and ICU staff. However, it is important to note that the majority of weaning decisions followed the Ventilator Weaning Program published by the Japanese Society of Intensive Care Medicine. Part of the description and a citation of these guidelines are provided below.

“Extubation is attempted in patients with successful SBT; the five criteria for the initiation of SBT are as follows: 1)Adequate oxygenation, defined as SpO2 >90% when FIO2 is <0.5 and PEEP <8 cmH2O. 2)Hemodynamic stability, with no acute myocardial ischemia or severe arrhythmia, a heart rate ≤140 bpm, and only small doses of vasoactive agents are employed; dopamine �5 μg/kg/min, dobutamine �5 μg/kg/min, noradrenaline �0.05 μg/kg/min. 3) Adequate ventilation effort, with ventilation volume >5 ml/kg per breath, minute ventilation <15 L/min, and absence of respiratory acidosis (pH >7.25). 4) No abnormal breathing patterns, excessive use of respiratory support muscles, or paradoxical breathing. 5) General condition is stable with no fever, severe electrolyte imbalances, severe anemia, or severe fluid overload. The success criteria for SBT were defined as the following conditions being met: 1) Respiratory rate <30 breaths/min. 2) No significant decline compared to before SBT starting, such as SpO2 ≥94% or PaO2 ≥70 mmHg. 3) Heart rate <140 bpm with no evidence of new arrhythmia or myocardial ischemia. 4) No excessive increase in blood pressure.”

“The Japanese Society of Intensive Care Medicine, the ventilator weaning protocol, https://www.jsicm.org/pdf/kokyuki_ridatsu1503a.pdf”

Although there was no pre-defined weaning protocol, the following description on the weaning method employed in the present study was added to the Methods section.

“Daily assessments of the ability to breathe with pressure support ventilation were performed when the positive end-expiratory pressure and fraction of inspiratory oxygen levels were lower than the day before. Additionally, the ventilator was switched to pressure support ventilation if the attending physician considered the patient sufficiently awake to breathe with pressure support ventilation. Extubation was attempted in patients with a successful spontaneous breathing trial (SBT). The ICU attending physician made the final decision regarding whether to extubate the patient.” (line 122-129)

Furthermore, in accordance with the Reviewer’s comment, the following sentences on this point were added to the Discussion section.

“Since 3blinding was not possible due to the nature of the intervention, there may have been performance, reporting, and selection biases. Suctioning may be performed more effectively and carefully in the MI-E group. MI-E increased the amount of secretions, which may have led the attending physician to delay extubation due to excessive secretions.” (line 330-335)

Comment 5: What is the tracheostomy policy? Is it early systematic tracheostomy, or late tracheostomy in selected patients?

Response: We thank the Reviewer for this comment. We did not actively perform early tracheostomy. Tracheostomy was performed when endotracheal intubation was anticipated to exceed 14 days. Therefore, the following sentence was added to the Methods section.

“Tracheostomy was performed if the ventilator duration was expected to be more than 14 days.” (line 129-130)

The following information was added to the Results section on the actual duration from intubation to tracheostomy.

“The median duration from intubation to tracheostomy was 12 (IQR, 9–19) in the MI-E group and 12 (IQR, 6–15) in the control group; P=0.85.” (line 228-229)

Comment 6: Who performed the mechanical insufflation-exsufflation? Please detail the protocol (the sentence “mechanical insufflation-exsufflation was used 10 time for each sputum aspiration”, lines 112-113 is not clear), including its weaning (line 116 is not clear).

Response: We wish to thank the Reviewer for this comment. Mechanical insufflation-exsufflation was performed by the nurse in charge who fully understood the protocol of the study and had received adequate training and practical experience with MI-E. In accordance with the Reviewer’s comment, we changed sentences on the MI-E protocol and its weaning from

“In both groups, subjects who required suctioning at least once every hour at the start of the study were included in the research. However, if they no longer needed to suction frequently based on the clinical assessment after study inclusion, they were allowed to suction less often than once every hour.”

to

“Suctioning was performed on-demand by the nurse in charge who fully understood the protocol of the study and had received adequate training and practical experience with suctioning. Although subjects who required suctioning at least once every hour at the start of the study were included in the research, there was no upper limit to the number of times. Moreover, if it was not possible to suction airway secretions after inclusion and no further suctioning of airway secretions was expected to be possible based on a physical examination and auscultation, suctioning was allowed to occur less frequently than once an hour.” (line 131-139)

and

“In the MI-E group, if there were no more airway secretions to be cleared even after using MI-E, suctioning and the use of MI-E were permitted at intervals exceeding one hour.” (line 146-148)

Comment 7: The secondary outcomes should be listed explicitly.

Response: We wish to thank the Reviewer for this comment. We apologize for the unclear method of presentation, which was edited in the revised manuscript by a native English speaker as follows.

“Secondary outcomes were the duration of mechanical ventilation up to day 28, the ICU length of stay, the mortality rate, and the tracheostomy rate. Secondary outcomes, except for mortality, were analyzed exclusively for patients who survived.” (line 166-169)

Comment 8: Respectfully, “the ability to remain free from mechanical ventilation for 24 hours” is not the same between extubation and tracheostomy (lines 130-132). Airway suctioning is easier in tracheostomized patients than in extubated patients.

Response: We wish to express our deep appreciation to the Reviewer for their insightful comment on this point, with which we agree. In the present study, it was necessary to aggregate the duration after extubation and after tracheostomy without mechanical ventilation through some method. A method that differentiates and aggregates the duration after extubation and after tracheostomy without mechanical ventilation separately would be ideal. However, to combine them, there was no alternative but to treat them equivalently. Since the points raised by the Reviewer are valid, the following sentences were added as a limitation in the Discussion section.

“In the present study, ventilator-free days for patients following extubation and tracheotomy without mechanical ventilation were counted as equivalent. While tracheotomized patients are more likely to clear airway secretions than extubated patients, there is no feasible method to unify or convert ventilator-free days between extubated and tracheotomized patients. Therefore, we were compelled to treat ventilator-free days in extubated and tracheotomized patients as equivalent.” (line 346-352)

Comment 9: Please detail the adverse events reviewed.

Response: We thank the Reviewer for this comment. We agree that more detailed descriptions of both potential adverse events and the actual adverse events that occurred are needed. In accordance with the Reviewer’s comment, we changed the sentence from

“There were no adverse events associated with this study.”

to

“While some patients had hemodynamic instability or arrhythmia at the time of inclusion, no new cases of hemodynamic instability or arrhythmia occurred with the use of MI-E. The use of MI-E did not lead to new cases of lung injury due to barotrauma, volutrauma, atelectrauma, or biotrauma. No oth

---

## [Decision Letter · Decision Letter 1]

1 Dec 2023

PONE-D-23-25920R1Effects of mechanical insufflation-exsufflation on ventilator-free days in intensive care unit subjects with sputum retention; A randomized clinical trialPLOS ONE

Dear Dr. Nakamura,

Thank you for submitting your manuscript to PLOS ONE. After careful consideration, we feel that it has merit but does not fully meet PLOS ONE’s publication criteria as it currently stands. Therefore, we invite you to submit a revised version of the manuscript that addresses the points raised during the review process.

We look forward to receiving your revised manuscript.

Kind regards,

Jean Baptiste Lascarrou

Academic Editor

PLOS ONE

Journal Requirements:

Reviewers' comments:

Reviewer's Responses to Questions

**Comments to the Author**

1. If the authors have adequately addressed your comments raised in a previous round of review and you feel that this manuscript is now acceptable for publication, you may indicate that here to bypass the “Comments to the Author” section, enter your conflict of interest statement in the “Confidential to Editor” section, and submit your "Accept" recommendation.

Reviewer #2: (No Response)

2. Is the manuscript technically sound, and do the data support the conclusions?

Reviewer #2: Partly

3. Has the statistical analysis been performed appropriately and rigorously? 

Reviewer #2: Yes

4. Have the authors made all data underlying the findings in their manuscript fully available?

Reviewer #2: Yes

5. Is the manuscript presented in an intelligible fashion and written in standard English?

Reviewer #2: Yes

6. Review Comments to the Author

Reviewer #2: Thank you for submitting a revised manuscript since the last time. I have a few comments on the revised manuscript.

1. The exclusion criteria do not contain pneumothorax, which is the main adverse event when using M I-E.

Response: We appreciate the Reviewer’s comment on this point. Patients who were considered to have underlying conditions that may be exacerbated by MI-E were excluded by the attending physician. The following sentence in the Methods section was changed from

“Patients who met the following criteria were excluded from the study: pregnant women, […], and cases that were deemed inappropriate by the attending physician.”

to

“Patients who met the following criteria were excluded from the study: pregnant women, […], and patients considered by the attending physician to have a coexisting condition that is exacerbated by MI-E, such as pneumothorax, empyema with fistula, or hemoptysis.” (line 96-101)

Comment: I accepted the authors' response .However, quality-assured clinical studies should include details of exclusion criteria in the protocol.

2. More than 80% of the subjects are male, but how many have a smoking history or chest C T that may have a bullae?　The MI-E pressure of 40 cm H2O was performed in all the same cases, but it is necessary to mention how it was managed and whether there were no adverse events.

Response: We appreciate the Reviewer’s comment on this point. In the present study, the collection of smoking history and chest CT scans were not specified in the study protocol. However, at Hitachi General Hospital, routine smoking history inquiries are conducted upon admission. Therefore, we retrospectively examined the smoking history of all patients. Additionally, since all patients included in the present study had undergone chest CT imaging, we retrospectively assessed the percentage of patients with lung bullae. The results obtained showed that 33% of patients in the control group and 48% in the MI-E group had a history of smoking, while 62% in the control group and 59% in the MI-E group had bullae in the lungs.

We also provided more detailed information on how to connect the MI-E machine and adjust pressure settings in the Methods section.

“When the MI-E COMFORT COUGH-Ⅱ is directly connected to the endotracheal tube and the pressure is set to 40 cmH2O, it applies both positive and negative pressures of 40 cmH2O during inhalation and exhalation, respectively, inside the endotracheal tube.” (line 141-144)

More detailed descriptions of actual adverse events were also added to the Results section.

“While some patients had hemodynamic instability or arrhythmia at the time of inclusion, no new cases of hemodynamic instability or arrhythmia occurred with the use of MI-E. The use of MI-E did not lead to new cases of lung injury due to barotrauma, volutrauma, atelectrauma, or biotrauma. No other apparent adverse effects attributed to MI-E occurred in the present study, such as pneumothorax, hemoptysis, oxygen desaturation, and mucus plugging.” (line 232-237)

Based on previous studies, the following sentences on the adverse event of pneumothorax, in connection with a smoking history or bullae were added to the Discussion section.

“In 13 studies that used MI-E, 10 found no adverse events, while 3 reported oxygen desaturation, hemodynamic variation, pneumothorax, mucus plugging, hemoptysis, and chest pain as adverse events.[17] In the present study, 33% of patients in the control group and 48% in the MI-E group had a history of smoking. All patients had undergone chest CT, and the percentage of patients with bullae in the lungs was 62% in the control group and 59% in the MI-E group. Pneumothorax did not occur in patients, even in those with a history of smoking or bullae.” (line 302-309)

Comment: I understood authors response. If the author mentions a history of smoking or bullae in the discussion, it should be described in the results.

In the MI-E group, 59% of patients had pulmonary bullae but did not experience any adverse events. This is an important result for the risk of MI-E.

The maximum size of the bullae is an important information with regard to the risk of MI-E.

3. In line 106 of the Randomisation and procedures, the word "respiratory chest physiotherapy" are not often used and should be corrected to either "chest physiotherapy" or "respiratory physiotherapy".

response: We thank the Reviewer for this comment. This expression was standardized to “respiratory physiotherapy”.

Comment: I accepted the authors' response.

4. Lines 227-229; the reference to 40 mmHg is cited as justification for setting MI-E pressure in this study, but the unit of 40 mmHg is 54 cm H2O when calculated in cm H2O, which may mislead the reader. Reference: https://pubmed.ncbi.nlm.nih.gov/25492956/

Response: We strongly agree with the Reviewer’s comment. We apologize for any confusion. Units need to be consistently aligned and were standardized to cmH2O. The following changes were made in the description from

“Pressures of 40 mmHg to –40 mmHg are usually most effective and best tolerated.”

to

“Since another study demonstrated that pressures of ±54 cmH2O were tolerated well, we assumed that a pressure of ±40 cmH2O may also be tolerated.” (line 298-299)

Comment: Reference (21) needs to be checked, as it does not mention the sentence "since another study showed that a pressure of ±54 cmH2O is well tolerated". The reference to be cited is "DOI:" ext-link-type="uri" xlink:type="simple">https://doi.org/10.4187/respcare.03584", which has been　previously noted.

5. The structure of the discussion lacks logic.

It would be better to describe what was expected in this hypothesis, what were the results, what were the factors, sample size and other factors due to the study design have been mentioned, but are there other factors, are there any respiratory physiological factors, and should include reference to medical factors.

Response: We wish to express our deep appreciation to the Reviewer for their insightful comment on this point. The expected hypothesis and the results obtained were added to the beginning of the Discussion section as follows.

“The hypothesis of the present study was that the additional use of MI-E in conventional pulmonary care shortens the duration of mechanical ventilation in subjects with high airway secretions. However, the use of MI-E before each direct tracheal suction did not significantly reduce the number of ventilator-free days by day 28. In addition, no significant differences were observed in the duration of mechanical ventilation, the ICU length of stay, the mortality rate, or the tracheostomy rate between the MI-E and control groups.” (line 247-253)

In the present study, we considered several factors that may have contributed to the lack of significant differences. Even if there were improvements in airway resistance or lung compliance, we considered the possibility that fundamental reasons for requiring mechanical ventilation, such as pneumonia, cardiac arrest, or sepsis, were not improved. We also entertained the possibility that the effectiveness of MI-E may have been insufficient or that there may have been potential adverse events associated with its use. The following sentences were added to the Discussion section.

“While the use of MI-E may reduce airway resistance or enhance compliance [7,8], it does not directly address the underlying medical factors (such as pneumonia, cardiac arrest, sepsis, and surgery) that necessitated intubation. This may explain why the number of ventilator-free days was not significantly different. A pressure of ±40 cmH2O may have been insufficient for patients with excessive sputum production, such as those examined in the present study. On the other hand, although MI-E was effective, potential adverse events, such as barotrauma or atelectrauma, may have affected the results obtained.” (line 256-263)

Comment: Smoking is known to increase the amount of airway mucus.

It may be mentioned in the discussion that smoking history was higher in the MI-E group (48%) than in the control group (33%) may have affected the results.

In a discussion, the description needs to be more structured for ease of reading.

1. Summary of research hypotheses and results

2. Discussion of the efficacy of MI-E

3. Consideration of adverse events.

4. Limitations

In this manus, I think the above structure is used.

In the discussion of efficacy in 2,

Firstly, the sample size that is considered to have the most effect. The next structure is to describe the influence of smoking history, medical factors and MI-E pressure settings.

This structure would be easier for the reader to understand.

6. New comment

The order of the MI-E and control groups in Tables 1 and 2 is not aligned. They should be aligned in order to show the reader clearly.

7. PLOS authors have the option to publish the peer review history of their article (what does this mean?). If published, this will include your full peer review and any attached files.

Reviewer #2: No

---

## [Author Response · Author response to Decision Letter 1]

11 Jan 2024

Dear Dr. Lascarrou,

Thank you for your e-mail regarding our manuscript entitled “Effects of mechanical insufflation-exsufflation on ventilator-free days in intensive care unit subjects with sputum retention; A randomized clinical trial”, and the valuable comments of the Reviewer. Please find attached our revised manuscript as well as point-by-point responses to the Reviewer’s comments. 

As indicated by the Reviewer, there was an error in one of the references. We sincerely apologize for the incorrect citation. We changed the reference in the sentence “Since another study demonstrated that pressures of ±54 cmH2O were tolerated well” (line 306-307) from

“Bach JR, Mehta AD. Respiratory muscle aids to avert respiratory failure and tracheostomy: a new patient management paradigm. Journal of Neurorestoratology. 2014;2:25-35.”

to

“Bach JR, Sinquee DM, Saporito LR, Botticello AL. Efficacy of mechanical insufflation-exsufflation in extubating unweanable subjects with restrictive pulmonary disorders. Respir Care. 2015 Apr;60(4):477-83.”

We consider the revised manuscript to be a suitable response to the Reviewer’s comments and it has been significantly improved over the previous submission. We trust that it is now suitable for publication in PLOS ONE. Thank you in advance for your kind consideration of our manuscript.

Sincerely yours,

Shota Kubota

Department of Emergency and Critical Care Medicine, Hitachi General Hospital, Ibaraki, Japan

Comment 1. The exclusion criteria do not contain pneumothorax, which is the main adverse event when using MI-E.

Previous Response: We appreciate the Reviewer’s comment on this point. Patients who were considered to have underlying conditions that may be exacerbated by MI-E were excluded by the attending physician. The following sentence in the Methods section was changed from

“Patients who met the following criteria were excluded from the study: pregnant women, […], and cases that were deemed inappropriate by the attending physician.”

to

“Patients who met the following criteria were excluded from the study: pregnant women, […], and patients considered by the attending physician to have a coexisting condition that is exacerbated by MI-E, such as pneumothorax, empyema with fistula, or hemoptysis.” (line 96-101)

Comment: I accepted the authors' response .However, quality-assured clinical studies should include details of exclusion criteria in the protocol.

Response: We wish to thank the Reviewer for this comment, with which we agree. 

We apologize for not providing a detailed definition of exclusion criteria in the study protocol. The following sentences were added to the Discussion section.

“Regarding the exclusion criteria of the present study, patients considered by the attending physician to have a coexisting condition exacerbated by MI-E, such as pneumothorax, empyema with fistula, or hemoptysis, were excluded. However, since the study protocol did not provide a detailed definition of exclusion criteria, there was a potential for selection bias.” (line 353-357)

Comment 2. More than 80% of the subjects are male, but how many have a smoking history or chest C T that may have a bullae? The MI-E pressure of 40 cm H2O was performed in all the same cases, but it is necessary to mention how it was managed and whether there were no adverse events.

Previous Response: We appreciate the Reviewer’s comment on this point. In the present study, the collection of smoking history and chest CT scans were not specified in the study protocol. However, at Hitachi General Hospital, routine smoking history inquiries are conducted upon admission. Therefore, we retrospectively examined the smoking history of all patients. Additionally, since all patients included in the present study had undergone chest CT imaging, we retrospectively assessed the percentage of patients with lung bullae. The results obtained showed that 33% of patients in the control group and 48% in the MI-E group had a history of smoking, while 62% in the control group and 59% in the MI-E group had bullae in the lungs.

We also provided more detailed information on how to connect the MI-E machine and adjust pressure settings in the Methods section.

“When the MI-E COMFORT COUGH-II is directly connected to the endotracheal tube and the pressure is set to 40 cmH2O, it applies both positive and negative pressures of 40 cmH2O during inhalation and exhalation, respectively, inside the endotracheal tube.” (line 141-144)

More detailed descriptions of actual adverse events were also added to the Results section.

“While some patients had hemodynamic instability or arrhythmia at the time of inclusion, no new cases of hemodynamic instability or arrhythmia occurred with the use of MI-E. Furthermore, the use of MI-E did not lead to new cases of lung injury due to barotrauma, volutrauma, atelectrauma, or biotrauma. No other apparent adverse effects attributed to MI-E occurred in the present study, such as pneumothorax, hemoptysis, oxygen desaturation, and mucus plugging.” (line 232-237)

Based on previous studies, the following sentences on the adverse event of pneumothorax, in connection with a smoking history or bullae were added to the Discussion section.

“In 13 studies that used MI-E, 10 found no adverse events, while 3 reported oxygen desaturation, hemodynamic variation, pneumothorax, mucus plugging, hemoptysis, and chest pain as adverse events. In the present study, 33% of patients in the control group and 48% in the MI-E group had a history of smoking. All patients had undergone chest CT, and the percentage of patients with bullae in the lungs was 62% in the control group and 59% in the MI-E group. Pneumothorax did not occur in patients, even in those with a history of smoking or bullae.” (line 302-309)

Comment: I understood authors response. If the author mentions a history of smoking or bullae in the discussion, it should be described in the results.

In the MI-E group, 59% of patients had pulmonary bullae but did not experience any adverse events. This is an important result for the risk of MI-E.

The maximum size of the bullae is an important information with regard to the risk of MI-E.

Response: We strongly appreciate the Reviewer’s comment on this point. We added the percentage of patients with a history of smoking and bullae, and the maximum size of bullae to Table 1 in the Results section. We also revised the description on adverse events in the Discussion section with the following text.

“In the present study, 33% of patients in the control group and 48% in the MI-E group had a history of smoking. All patients had undergone chest CT, and the percentage of patients with bullae in the lungs was 62% in the control group and 59% in the MI-E group. The median of the maximum size of bullae was 12 mm (IQR, 5-17) in the control group and 12 mm (IQR, 7-30) in the MI-E group. Pneumothorax did not occur in patients, even in those with a history of smoking or bullae.” (line 311-318)

Comment 3. Reference (21) needs to be checked, as it does not mention the sentence "since another study showed that a pressure of ±54 cmH2O is well tolerated". The reference to be cited is "DOI:https://doi.org/10.4187/respcare.03584", which has been previously noted.

Response: We appreciate the Reviewer’s comment on this point, with which we agree. We sincerely apologize for the incorrect citation of references. We changed the reference in the sentence “Since another study demonstrated that pressures of ±54 cmH2O were tolerated well” (line 306-307) from

“Bach JR, Mehta AD. Respiratory muscle aids to avert respiratory failure and tracheostomy: a new patient management paradigm. Journal of Neurorestoratology. 2014;2:25-35.”

to

“Bach JR, Sinquee DM, Saporito LR, Botticello AL. Efficacy of mechanical insufflation-exsufflation in extubating unweanable subjects with restrictive pulmonary disorders. Respir Care. 2015 Apr;60(4):477-83.”

Comment 4: Smoking is known to increase the amount of airway mucus. It may be mentioned in the discussion that smoking history was higher in the MI-E group (48%) than in the control group (33%) may have affected the results.

In a discussion, the description needs to be more structured for ease of reading.

1. Summary of research hypotheses and results

2. Discussion of the efficacy of MI-E

3. Consideration of adverse events.

4. Limitations

In this manus, I think the above structure is used.

In the discussion of efficacy in 2,

Firstly, the sample size that is considered to have the most effect. The next structure is to describe the influence of smoking history, medical factors and MI-E pressure settings.

This structure would be easier for the reader to understand.

Response: We wish to express our deep appreciation to the Reviewer for their insightful comment on this point. We added the following text to 2 ("Reasons for the absence of significant findings") to make it easier for the reader to understand.

“Several factors may account for the observed lack of a significant increase in the number of ventilator-free days with MI-E. The primary contributing factor is the limited sample size. Prior to the initiation of the present study, the target difference in ventilator-free days between the two groups was designed as 4 days and the sample size was set at 60 subjects. We planned to terminate enrollment once the target sample size was reached or in June 2019. Fewer ICU-ventilated subjects met the inclusion criteria than expected, and, thus, the study was terminated before the target sample size was reached. The insufficient sample size may have affected the power of the study. Therefore, randomized open-label trials with larger sample sizes are needed. Furthermore, the percentage of patients with a history of smoking may have affected the results. In the present study, 33% of patients in the control group and 48% in the MI-E group had a history of smoking. Since smoking is known to increase the amount of airway mucus [13], the higher percentage of patients with a smoking history in the MI-E group than in the control group may also have affected the results. In addition, medical factors may have contributed to the lack of a significant difference. In patients without neuromuscular disease, even if airway secretions are high, the ability to exhale may be sufficient without MI-E. While the use of MI-E may reduce airway resistance or enhance compliance [7,8], it does not directly address the underlying medical factors (such as pneumonia, cardiac arrest, sepsis, or surgery) that necessitated intubation. This may explain why the number of ventilator-free days was not significantly different. Finally, the possibility that the pressure settings for MI-E were inappropriate may be a reason for the lack of a significant difference. A pressure of ±40 cmH2O may have been insufficient for patients with excessive sputum production, such as those examined in the present study. On the other hand, although MI-E was effective, potential adverse events, such as barotrauma or atelectrauma, may have affected the results obtained.” (line 254-278)

I hope that the modified structure, along with the additional sentences, has enhanced readability for the readers.

Comment 5: The order of the MI-E and control groups in Tables 1 and 2 is not aligned. They should be aligned in order to show the reader clearly.

Response: We wish to thank the Reviewer for this comment. We reversed the order of the control group and MI-E group in Table 2 to match that in Table 1.

We wish to thank the Reviewer again for their valuable comments.

---

## [Decision Letter · Decision Letter 2]

7 Feb 2024

PONE-D-23-25920R2Effects of mechanical insufflation-exsufflation on ventilator-free days in intensive care unit subjects with sputum retention; A randomized clinical trialPLOS ONE

Dear Dr. Nakamura,

Thank you for submitting your manuscript to PLOS ONE. After careful consideration, we feel that it has merit but does not fully meet PLOS ONE’s publication criteria as it currently stands. Therefore, we invite you to submit a revised version of the manuscript that addresses the points raised during the review process.

We look forward to receiving your revised manuscript.

Kind regards,

Jean Baptiste Lascarrou

Academic Editor

PLOS ONE

Journal Requirements:

Reviewers' comments:

Reviewer's Responses to Questions

**Comments to the Author**

1. If the authors have adequately addressed your comments raised in a previous round of review and you feel that this manuscript is now acceptable for publication, you may indicate that here to bypass the “Comments to the Author” section, enter your conflict of interest statement in the “Confidential to Editor” section, and submit your "Accept" recommendation.

Reviewer #2: All comments have been addressed

Reviewer #4: All comments have been addressed

2. Is the manuscript technically sound, and do the data support the conclusions?

Reviewer #2: Yes

Reviewer #4: Yes

3. Has the statistical analysis been performed appropriately and rigorously? 

Reviewer #2: Yes

Reviewer #4: Yes

4. Have the authors made all data underlying the findings in their manuscript fully available?

Reviewer #2: Yes

Reviewer #4: Yes

5. Is the manuscript presented in an intelligible fashion and written in standard English?

Reviewer #2: Yes

Reviewer #4: Yes

6. Review Comments to the Author

Reviewer #2: I accepted your revised manuscript. I really appreciate your work and efforts. I hope this study will be read by many people and contribute to the development of the intensive care field.

Reviewer #4: The manuscript is clearly written.

Very minor comments, which will add to the transparency of reporting.

1. For randomisation; no block used, therefore assumption is that, this was a simple randomisation - authors include this in the manuscript.

2. This was an open-label study, when was the analysis plan finalized, ideally prior to first participant recruited.

3. In statistical analysis section, mention how missing data would be handled if at all?

4. Define your population of analysis (e.g analysed as randomised irrespective of whether they received the intervention or not).

5. Table 1- not recommended to test for baseline characteristics as this an RCT, any differences observed would be due to chance - omit p-values from table 1 as mis-leading.

6. Such a wide gap of age, it would help if you can present the age-category into sensible categories, to understand the distribution of who is included in your sample?

7. For descriptions purposes, also include a sensible cut-off of the primary outcome, i.e curious that you have less days in the control group - hence why possible interest in age categories, i.e could be because they may be more younger people in control group that intervention group - influencing ventilator free days.

7. PLOS authors have the option to publish the peer review history of their article (what does this mean?). If published, this will include your full peer review and any attached files.

Reviewer #2: No

Reviewer #4: No

---

## [Author Response · Author response to Decision Letter 2]

15 Mar 2024

We wish to express our strong appreciation to the Reviewer for their insightful comments, which have helped us to significantly improve the quality of the revised manuscript.

Comment 1. For randomisation; no block used, therefore assumption is that, this was a simple randomisation - authors include this in the manuscript.

Response: We wish to thank the Reviewer for this comment. The following sentence in the Methods section was changed from

“The randomization process did not include random blocking.”

to

“Simple randomization was implemented and did not include random blocking.” (lines 117-118)

The following sentence in the Discussion section was changed from

“The number of patients in both groups was not 1:1, which may have affected the results obtained because the randomization process did not include random blocking.”

to

“The number of patients in both groups was not 1:1, which may have affected the results obtained because simple randomization was implemented and did not include random blocking.” (lines 361-363)

Comment 2. This was an open-label study, when was the analysis plan finalized, ideally prior to first participant recruited.

Response: We appreciate the Reviewer’s comment on this point. The following sentence was added to the Methods section.

“The analysis plan was finalized on 16th October 2017, prior to the recruitment of the first participant on 1st November 2017.” (lines 189-191)

Comment 3. In statistical analysis section, mention how missing data would be handled if at all?

Response: We thank the Reviewer for this comment. The following sentence was added to the Results section. 

“There were no missing data.” (line 209)

Comment 4. Define your population of analysis (e.g analysed as randomised irrespective of whether they received the intervention or not).

Response: We appreciate the Reviewer’s comment on this point. The following sentence was added to the Methods section.

“In the present study, we performed an intention-to-treat analysis.” (line 191)

Comment 5. Table 1- not recommended to test for baseline characteristics as this an RCT, any differences observed would be due to chance - omit p-values from table 1 as mis-leading.

Response: We strongly appreciate the Reviewer’s comment on this point. We omitted p-values from table 1.

Comment 6. Such a wide gap of age, it would help if you can present the age-category into sensible categories, to understand the distribution of who is included in your sample?

Response: We wish to express our deep appreciation to the Reviewer for the insightful comment on this point. We added age categories to Table 1. Patients in the Control group and MI-E group were categorized into three age groups: 60 years and younger, between 61 and 80 years, and 81 years and older.

Comment 7. For descriptions purposes, also include a sensible cut-off of the primary outcome, i.e curious that you have less days in the control group - hence why possible interest in age categories, i.e could be because they may be more younger people in control group that intervention group - influencing ventilator free days.

Response: We strongly appreciate the Reviewer’s comment on this point. Since the difference in age distribution among patients in the Control group and MI-E group may have potentially affected the results obtained, the following sentence was added to the Discussion section.

“Patients in the control group were younger than those in the MI-E group, which may have affected the results obtained.” (lines 269-271)

Based on the present results on the duration of mechanical ventilation, the reduction of four days for sample size estimations may have been too substantial. The following sentences were added to the Discussion section.

“In the present study, the sample size was calculated to detect a reduction of 4 ventilator-free days in the MI-E group for the primary endpoint. Since the present results showed that the duration of mechanical ventilation was 6 in the MI-E group and 8 in the control group, setting the cut-off of the primary outcome at 4 days may have been too large.” (lines 363-367)

We wish to thank the Reviewer again for their valuable comments.

---

## [Editor Report · Decision Letter 3]

1 Apr 2024

Effects of mechanical insufflation-exsufflation on ventilator-free days in intensive care unit subjects with sputum retention; A randomized clinical trial

PONE-D-23-25920R3

Dear Dr. Nakamura,

We’re pleased to inform you that your manuscript has been judged scientifically suitable for publication and will be formally accepted for publication once it meets all outstanding technical requirements.

Kind regards,

Jean Baptiste Lascarrou

Academic Editor

PLOS ONE